# Climate Change Promotes the Large-Scale Population Growth of *Grapholita molesta* (Busck) (Lepidoptera: Tortricidae) within Peach Orchards in China

Hongchen Li [1,2,†], Qiulian Peng [3,†], Su Wang [2], Fan Zhang [2], Xiaojun Guo [4], Quan Jiang [5], Ningxing Huang [2,*] and Hu Li [1,*]

1   MOA Key Lab of Pest Monitoring and Green Management, Department of Entomology, College of Plant Protection, China Agricultural University, Beijing 100193, China
2   Institute of Plant Protection, Beijing Academy of Agriculture and Forestry Sciences, Beijing 100097, China
3   Yunan Academy of Agricultural Sciences, Kunming 650205, China
4   Beijing Academy of Agriculture and Forestry Sciences, Beijing 100097, China
5   Beijing Academy of Forestry and Pomology Sciences, Beijing Academy of Agriculture and Forestry Sciences, Beijing 100097, China
*   Correspondence: huangningxing@ipepbaafs.cn (N.H.); tigerleecau@hotmail.com (H.L.)
†   These authors contributed equally to this work.

**Abstract:** Cosmopolitan agricultural herbivorous pests are provided with a wide range of potential hosts. Therefore, they have high carrying capacity, and can cause extremely severe damage in agroecosystems. Understanding the ecological mechanisms of their population dynamics, especially as they relate to large-scale meteorological variations and geographical landscape influences, can help us to reveal how they became such important pests. The oriental fruit moth, *Grapholita molesta*, is a typical example of a significant pest distributed on a large scale, which is capable of damaging fruit trees of economic value such as peach, apple, pear, etc. This pest not only occurs in China, but exists on all continents except Antarctica. In order to prevent major pests and diseases, a system of plant protection has been established gradually in peach orchards within the Modern Agro-industry Technology Research System in China (CARS) since 2009. In the system, we collected the monitoring data of *G. molesta* by using pheromone traps at 17 experimental stations, and then used the corresponding climate data (temperature and precipitation) to explore the link between climate factors using mixed models. The results show that both monthly mean temperature and precipitation had a significant positive correlation with the occurrence of *G. molesta*. Therefore, global warming with higher levels of precipitation may favor *G. molesta*, allowing it to outperform other potential pests at the population level in peach orchards, on a large scale.

**Keywords:** *Grapholita molesta*; population dynamics; climate change

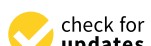



## 1. Introduction

The weather is becoming warmer [1–4], and climate change is affecting the Earth faster than scientists predicted [5]. Ecological responses to climate change are visible and are associated with a temperature increase [6], with temperature being the greatest single factor affecting the geographic distribution of any species of insect [7]. The potential rate of increase in many insects is also strongly dependent on temperature [8], which has been identified as the dominant abiotic factor directly affecting herbivorous insects [1]. Therefore, considering insect populations, climate change has a variety of effects. These include changing the abundance and the magnitude of pest outbreaks [9] and profoundly affecting the status and population dynamics of agricultural insect pests [10]. Although it is well known that insects are sensitive to temperature, how they will be affected by ongoing global warming remains uncertain due to the ecological complexity of the situation [11].

Insects, as poikilotherms, will experience stronger and more obvious changes to their ecology and adaptability to climate change, including their distribution area [12], population dynamics [13], and phenological changes [14]. A small change in climate may have many great impacts on insects; therefore, insects have a high risk of proliferation and invasion. They cause billions of dollars in direct agricultural losses through reduced yields and the transmission of plant pathogens, as well as indirect losses, including the increased application of pesticides and trade restrictions [15]. The increase in the level of intercontinental trade activities has made it easier for insects to proliferate and invade. Through cross-continent transmission, many insects tend to gradually reproduce and develop into worldwide large-scale pests with high adaptability after entering the local climate environment. The particular climate experienced by insects is a result of a mix of factors operating on diverse spatial scales, with variation in the relative importance of each of these scales [16]. In a 37-year field study of a worldwide pest, the cotton bollworm (*Helicoverpa armigera* (Hübner)), in China, the outbreak of the population was affected by both temperature and rainfall changes, along with changes in agricultural intensification, resulting in a population outbreak [17]. The workload of insect population prediction and model building has far exceeded the abilities of empirical research, which has caused researchers to ask whether we should continue collecting empirical data on insect responses to climate change or focus on higher-profile modeling, putting these model results toward conservation aims.

We consider empirical data on insect responses to climate change to be of vital importance. By analyzing the response of representative cosmopolitan major pests to climate change through empirical data, we can understand the reasons why major pests represent such a major hazard and what role climate factors play in this process. In the past 13 years, a system of field monitoring and study for pests in peach orchards, within the Modern Agro-industry Technology Research System in China (CARS), has been established in China to find solutions for managing these important pests. With the assistance of CARS, we collected the monitoring data of *Grapholita molesta* (Busck), the oriental fruit moth, which is a major pest in China and is widely distributed all over the world, except for in Antarctica [18]. It is a major pest of stone fruit species mainly belonging to the Rosaceae family, with a wide range of hosts, including peach, apple, pear, nectarine, cherry, quince, and persimmon (Ebenaceae) [19]. The ability of *G. molesta* to live in plant tissue for the long term increases the difficulty of effective control, and its wide range of hosts, its hidden living environment, and the unsatisfactory effect of chemical control have led to the oriental fruit moth rapidly representing a greater hazard and becoming a major pest in Rosaceae fruit tree planting areas.

Currently, the oriental fruit moth mainly occurs in temperate regions around the world, and occasionally in non-temperate zones, which may be due to the difference between the climate and the region caused by the local geographical environment [18,20–22]. Molecular analyses on the population level also showed that the moth, as an ecological generalist, has expanded its range globally during the past century. It has shown a pattern of genetic differentiation and an evolutionary history associated with geography [22], and the structure of its populations support the ecological strategies and evolutionary patterns that promote its successful expansion at a regional scale, alongside global climate warming [23]. Climate change models have also predicted that more areas will become suitable for the pest [21]. Previous biological or ecological studies on *G. molesta* have revealed the general trend of influence of climatic conditions on the development duration, population dynamics, and reproduction of this insect [21,24,25]. An appropriate increase in temperature within a certain range shortens the duration of development, whereas excessive precipitation reduces the population [23,26,27]. However, these previous studies often only provide the change trend, but do not demonstrate a clear and accurate change relationship at a large scale on the population level.

In brief, climate change may affect the population dynamics of *G. molesta*. However, knowledge of how these climate factors affect the pattern and the variability of the popu-

lation on a large scale, which is important for understanding the population ecology and evolution of this pest, is lacking. Therefore, based on the analysis of the monitoring data of *G. molesta*, we explored the relationship between climate factors and the occurrence of this pest. Warming-induced changes in the timing of emergence are well documented [28], so we focused on the changes in population dynamics. The objective of this research was to provide new evidence of how climate change affects the population of *G. molesta* in the agroecosystem on a large scale, and we also discussed the possible reasons for *G. molesta* becoming a dominant fruit pest in some regions of China.

## 2. Materials and Methods

### 2.1. Sampling of G. molesta

We collected the monitoring data of *G. molesta* by using pheromone traps from 17 experimental stations located within China's peach industry. The pheromone trap which was used most often was made of a sex pheromone lure (provided by Institute of Zoology, Chinese Academy of Sciences, from 2009 to now) and a plastic basin with a rainproof cover (provided by the plant protection system in the CARS). The basin was filled with water, with the lure near the water's surface. The experimental stations were distributed widely across peach-producing areas with incidences of *G. molesta*. The peach orchards where the data were collected were under approximately the same control pressure, with the same IPM protocols (adapted by local experimental stations according to different conditions) from the plant protection system in the CARS. We assumed that pest control affected the population of *G. molesta* with the same effects, for the purpose of simplifying the data analysis to focus on climate factors. The oriental fruit moths were monitored in peach orchards in 96 sites, belonging to 17 experimental stations, from 2009 to 2021 (Figure 1). These sampling sites were distributed along a 14.81° latitudinal gradient (from 25.00° N to 39.81° N, with longitudes from 102.65° E to 121.64° E; altitude range: 7.7–1893.4 m) extending from Kunming to Beijing (mean annual precipitation from 327 to 1888 mm, and mean temperature from 20.75 to 10.33 °C, with seasonality in precipitation), representing the four major regions of peach production in China and approximately 70~80 percent of the total peach production. Within each site, three to five sex pheromone traps were installed separately, with a distance of 25 m between replicates. The data were then grouped and standardized to provide the monthly sum number per trap caught at each site, as a proxy of the occurrence of *G. molesta*.

### 2.2. Climate Data

The climate data came from the daily meteorological element station observation data provided by the Resources and Environmental Science Data Center of the Chinese Academy of Sciences. The temperature data included daily average temperature, daily maximum temperature, and daily minimum temperature. The precipitation data included daily nighttime precipitation (20:00–8:00), daily daytime precipitation (8:00–20:00), and daily cumulative precipitation (20:00–20:00). Monthly mean temperature and precipitation (daily) were calculated based on the data above. Only data from within the period of monitoring were used for the analysis in the mixed model. Figure 2 shows the general trend of temperature for the 17 experimental stations within the past 13 years.

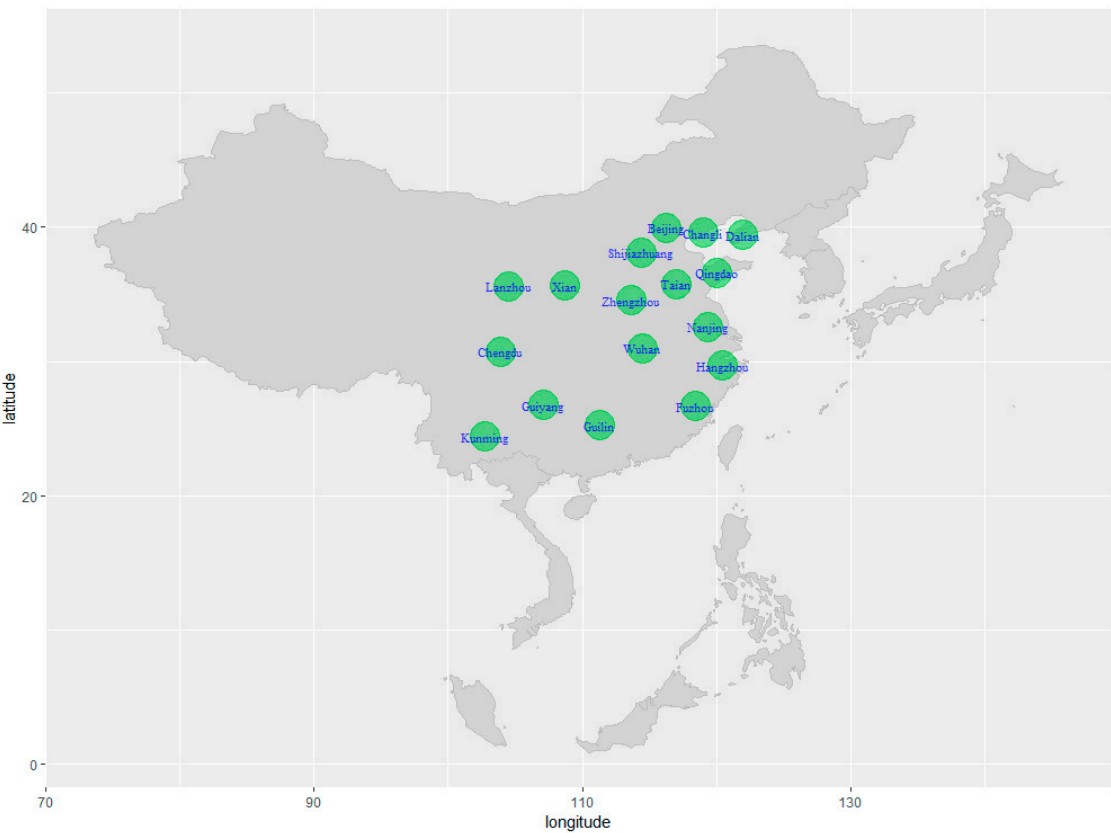

**Figure 1.** The position diagram of the 17 stations where *G. molesta* data were sampled (2009–2021).

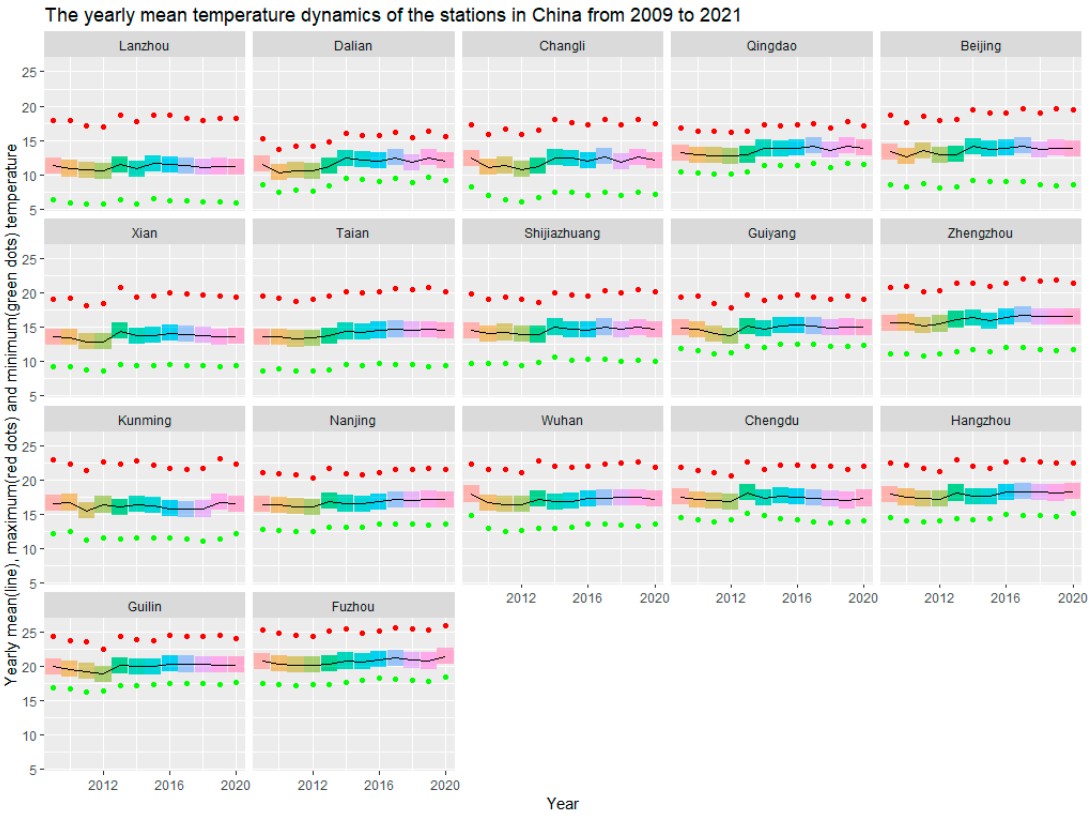

**Figure 2.** The yearly mean (line), maximum (red dots), and minimum (green dots) temperature dynamics of the 17 stations in China.

*2.3. Data Analysis*

The *G. molesta* data, i.e., the sum number per trap in one site within a month, were transformed to the natural logarithm as a response variable. They were then analyzed using linear mixed models with to monthly mean temperature and precipitation as the explanatory (predictor) variables, and a dataset with 1217 observations was created. We used the variance inflation factor (VIF) to conduct multicollinearity analysis on temperature and precipitation factors to find and eliminate the variables with strong collinearity at preliminary model selection. We selected the linear mixed model (more complicated models with non-normal error distributions, such as generalized linear mixed models with negative binomial distribution or Poisson distribution, etc., failed to better explain the data structure) to analyze the data, using the population dynamics of *G. molesta* as the response variable and the climate data as the predictor variables. The sites were nested in stations and months were nested in years as random factors to adjust the variability caused by the time series and the site locations.

All of the data analysis was completed in R 4.2.1 [29]. The "lmer" function in the "lme4" package [30] was used to fit the linear mixed model, and residual analysis was performed graphically to assess the model's assumptions. The "ggplot2" package [31] was used to create the figures. The Akaike information criterion (AIC) was used to sort all candidate models [32], and the general formula (in R syntax and "lme4") was as follows:

Models:

Null model:

$$\ln (G.\ molesta) \sim (1\,|\,\text{station}/\text{site}) + (1\,|\,\text{year}/\text{month})$$

Test models:

$$\ln (G.\ molesta) \sim \text{temperature} \times \text{precipitation} + (1\,|\,\text{station}/\text{site}) + (1\,|\,\text{year}/\text{month})$$

## 3. Results

*3.1. The Population Dynamics of G. molesta in China*

From 2009 to 2021, we collected and analyzed the monitoring data in 17 experimental stations in China. We found that the number of generations of *G. molesta* in China increased from north to south along the latitude (latitude by itself is not a determining factor, as it partly determines the temperature and precipitation), with two to four generations a year in North China and five to seven generations a year in South China. Additionally, the farther south the observation site was, the earlier the first occurrence time was (see Table 1).

**Table 1.** The generation and peak abundance of *G. molesta* in China.

| Region and Station | Generations | Time | Time of Peak Abundance |
|---|---|---|---|
| Northeast China (Dalian) | 2~4 | Early Apr to late Sep | Early to mid-May, late June, mid- to late July, late August |
| North China (Beijing, Changli, Qingdao, Shijiazhuang and Taian) | 3~5 | Early Apr to early Oct | Early to mid-May, late June, mid-July, mid-August, mid- to late September |
| Northwest China (Lanzhou and Xian) | 4~5 | Early Apr to early Oct | Early to mid-May, mid-June, early July, mid-August, mid-September |
| Central China (Zhengzhou and Wuhan) | 4~5 | Mid Mar to late Oct | Late March, late May, mid- to late June, late July, early September |
| Southwest China (Chengdu, Kunming and Guiyang) | 4~5 | Early Mar to mid Oct | Mid- to late May, mid- to late June, late July, early August, late August, early September, mid-September |
| East China (Hangzhou and Nanjing) | 5~6 | Late Mar to early Oct | Mid to late April, late May, late June, late July, late August, mid- to late September |
| South China (Fuzhou and Guilin) | 6~7 | Early Mar to late Oct | Early March, late April, mid-May, mid-June, mid-July, mid-August, mid- to late September |

### 3.2. Temperature and Precipitation Effects

Through the mixed model of climate variables and the study of *G. molesta* population dynamics, we obtained the following models with comparable parameters: the number of parameters, AIC, BIC, logLik, etc. (see Tables 2 and 3). All of the predictors in Table 2, except for monthly mean precipitation (day), had a significant positive effect on the response variable (the natural logarithm of *G. molesta*). Among the precipitation predictors, the monthly mean precipitation (during the night) was chosen for further screening based on the higher estimate of fixed effects (0.062; Type II Wald chi-square tests ($\chi^2 = 7.155$, $p < 0.01$, df = 1)), which indicates that it had more of an effect than the other precipitation predictors and the residual of the random effects. Table 3 shows that the models with predictor interactions between temperature and precipitation significantly improved the model, as indicated by the smaller value of the AIC. The model with monthly mean temperature and monthly mean precipitation (night) as predictors with interaction was the best among these six models, with a fixed effect estimate of 0.359 ($\chi^2 = 6.999$, $p < 0.01$, df = 1) for monthly mean precipitation (night) and 0.154 ($\chi^2 = 125.110$, $p < 0.001$, df = 1) for monthly mean temperature. Both values indicate that the monthly mean temperature and monthly mean precipitation (night) had a significant positive effect on *G. molesta* (Figure 3), with the interaction showing a slightly negative but significant effect ($-0.012$) ($\chi^2 = 6.412$, $p < 0.05$, df = 1). The interaction effect could be interpreted as the expected change in temperature effect for a one-unit change in precipitation, or vice versa. Although the model with the monthly mean maximum temperature and monthly mean precipitation (night) with interaction is not as good as the former, as the higher AIC value indicates, its biological meaning cannot be neglected. The upper threshold temperature of *G. molesta* was more connected to the warming trend than the other two temperature predictors (see Figure 4 for the lower and upper threshold effect; Figure 4 also showed that the effect of the monthly mean temperature (indicated by the slopes) on *G. molesta,* within the lower temperature range (slope = 0.6352, t = 5.771, $p < 0.001$, df = 61), was greater than within the higher temperature range (slope = $-0.1149$, t = $-1.112$, $p = 0.26778$, df = 147)). The model with the monthly mean maximum temperature and monthly mean precipitation (night) as predictors with interaction showed a fixed effect estimate of 0.489 ($\chi^2 = 12.266$, $p < 0.001$, df = 1) for mean precipitation (night), and 0.154 ($\chi^2 = 122.200$, $p < 0.001$, df = 1) for mean maximum temperature. Both values indicate positive effects, although the interaction showed a slightly negative significant effect ($-0.014$) ($\chi^2 = 8.252$, $p < 0.01$, df = 1). The interaction effect has the same meaning as mentioned above. Comparing these two models, we found that, when adjusting one predictor as well as the interaction, precipitation may have a greater effect on *G. molesta* than the temperature. For the convenience of explanation by biological meaning, the data were not transformed to the same scale.

**Table 2.** The models with a single predictor compared with the null model.

| Predictors | npar | AIC | BIC | logLik | Deviance | $\chi^2$ | Df | Pr (>$\chi^2$) |
|---|---|---|---|---|---|---|---|---|
| Null | 6 | 4434.4 | 4465.0 | −2211.2 | 4422.4 | | | |
| Mean temperature | 7 | 4335.9 | 4371.7 | −2161.0 | 4321.9 | 100.4 | 1.0 | <0.001 |
| Minimum temperature | 7 | 4335.3 | 4371.0 | −2160.6 | 4321.3 | 0.7 | 0.0 | |
| Maximum temperature | 7 | 4349.9 | 4385.6 | −2167.9 | 4335.9 | 0.0 | 0.0 | |
| Mean precipitation | 7 | 4431.2 | 4467.0 | −2208.6 | 4417.2 | 0.0 | 0.0 | |
| Mean night precipitation | 7 | 4429.5 | 4465.3 | −2207.8 | 4415.5 | 1.7 | 0.0 | |
| Mean day precipitation | 7 | 4434.6 | 4470.3 | −2210.3 | 4420.6 | 0.0 | 0.0 | |
| Monthly precipitation | 7 | 4430.9 | 4466.7 | −2208.5 | 4416.9 | 3.6 | 0.0 | |
| Monthly day precipitation | 7 | 4434.7 | 4470.5 | −2210.4 | 4420.7 | 0.0 | 0.0 | |
| Monthly night precipitation | 7 | 4430.6 | 4466.3 | −2208.3 | 4416.6 | 4.1 | 0.0 | |

**Table 3.** The models with two predictors and interaction compared with the models without it.

| Predictors | npar | AIC | BIC | logLik | Deviance | $\chi^2$ | Df | Pr (>$\chi^2$) |
|---|---|---|---|---|---|---|---|---|
| Mean temperature + mean night precipitation | 8 | 4331.2 | 4372.0 | −2157.6 | 4315.2 | | | |
| Maximum temperature + mean night precipitation | 8 | 4339.9 | 4380.8 | −2162.0 | 4323.9 | 0.0 | 0.0 | |
| Minimum temperature + mean night precipitation | 8 | 4334.2 | 4375.0 | −2159.1 | 4318.2 | 5.7 | 0.0 | |
| Mean temperature × mean night precipitation | 9 | 4326.8 | 4372.7 | −2154.4 | 4308.8 | 9.4 | 1.0 | $p < 0.01$ |
| Maximum temperature × mean night precipitation | 9 | 4333.7 | 4379.7 | −2157.9 | 4315.7 | 0.0 | 0.0 | |
| Minimum temperature × mean night precipitation | 9 | 4333.0 | 4378.9 | −2157.5 | 4315.0 | 0.7 | 0.0 | |

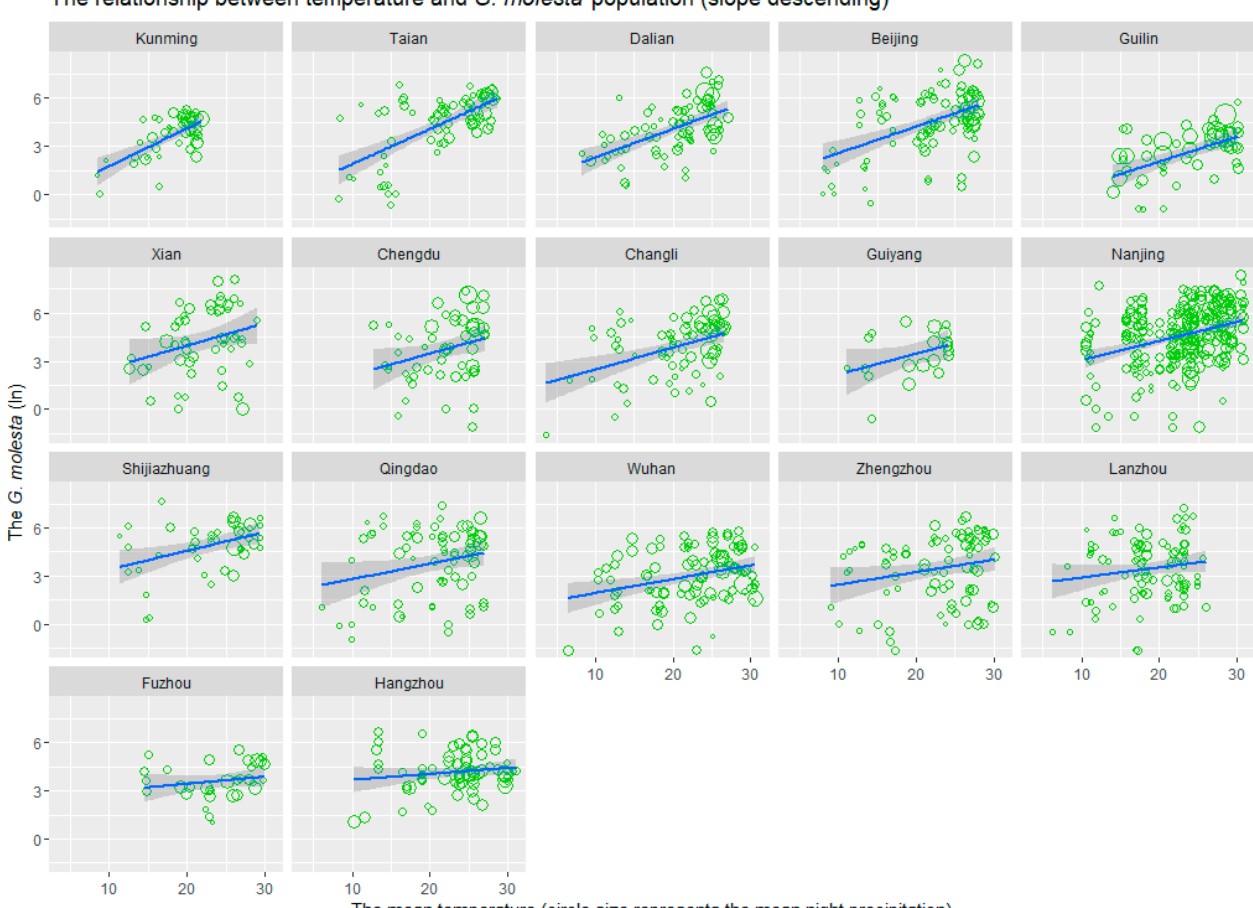

**Figure 3.** The relationship between mean temperature and *G. molesta* populations in 17 experimental sites in China.

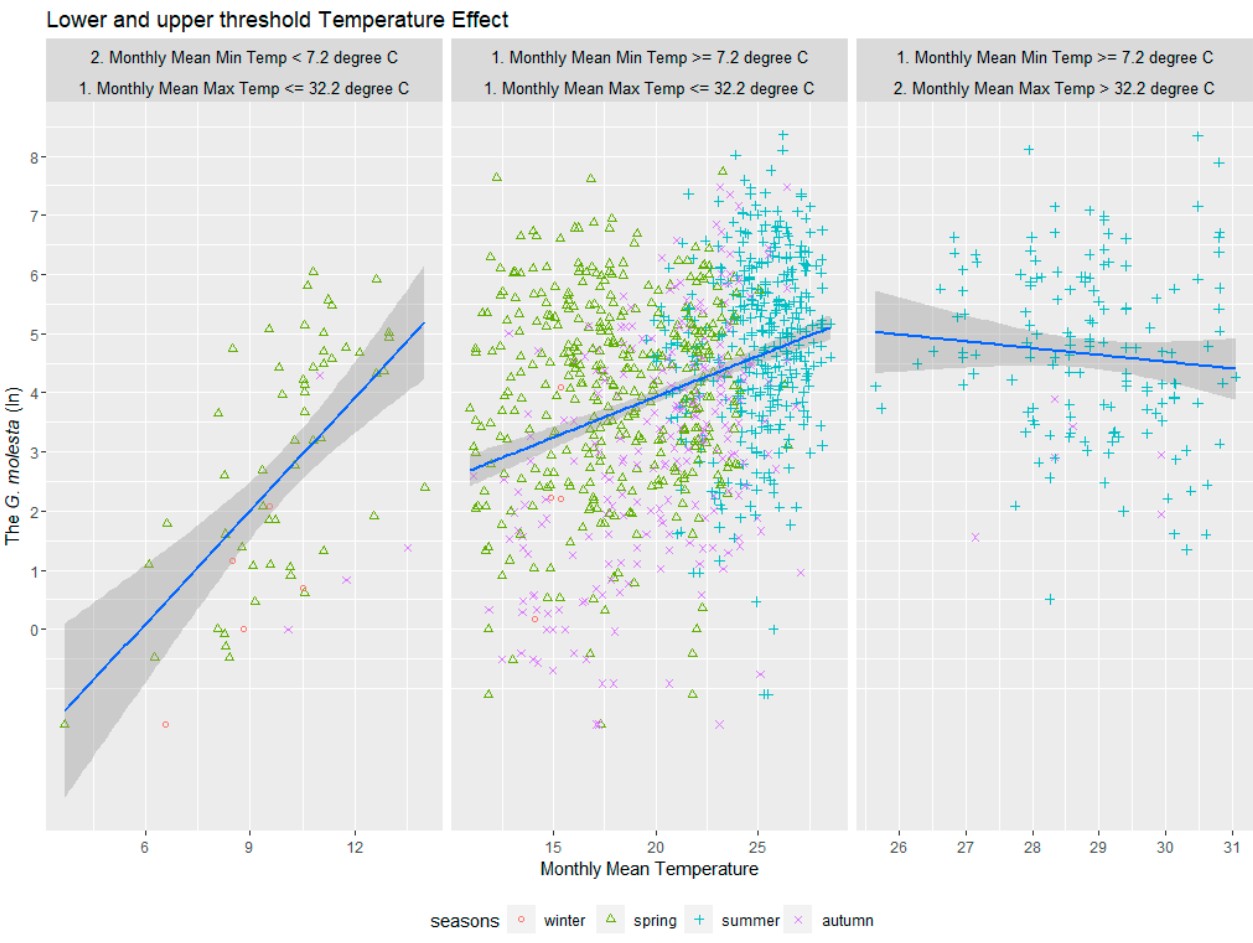

**Figure 4.** The lower and upper threshold effects on the relationship between temperature and *G. molesta* population from all sites of the 17 experimental stations.

## 4. Discussion

Changes in temperature are important for predicting the potential impact of global warming on pests; minimum and maximum temperature extremes can be particularly useful [6]. Our results show that when only one predictor was used in the models, the temperature parameters of the mean, minimum, and maximum all had a significantly positive effect on *G. molesta,* with similar fixed effects. In general, the parameter related to extreme temperature was supposed to have a greater effect on the population due to the limited temperature range for the development of this pest. Insect resistance to upper lethal temperatures is much less variable than their resistance to lower temperatures, and most insects are susceptible to heat stress between 28 °C and 32 °C, which is lower than commonly believed [33]. However, the data structure we collected had some limitations. For example, the population of *G. molesta* which we used in this study was not a direct field survey of the population; rather, it was a kind of indicator of the population dynamics based on the monitoring data of the adult males. The monitoring of *G. molesta* mainly started in March and ended in October, but during this period, the extreme low temperatures of the year were excluded. On the other hand, the maximum temperatures were included, but with the use of average data, the variability of the range was reduced. Furthermore, the duration of the extreme temperature for each day was not available; therefore, the model with maximum temperature as the predictor did not show any surprising information compared with the other two models. This result may indicate that *G. molesta* is not very sensitive to the maximum temperature at the population level on a large scale, or it may mean that *G. molesta* can endure the extreme temperatures at this stage. However, due to the limitation of our data, this assertion should be generalized cautiously outside of

the study areas, although the mixed model allows this kind of generalization with the study locations as the random effects. Analysis (Figure 4) of the effect of monthly mean temperature (indicated by the slopes) on *G. molesta* within the lower temperature range was greater than that within the higher temperature range, which also partly supported the hypothesis.

Some previous researches partly agreed with our results. For example, at low temperatures, with a beneficial acclimation effect, the most favorable time periods for female *G. molesta* dispersal are at the beginning and towards the end of the host crop's growing season [34]. In cooler environments, the offspring generation is expected to develop more rapidly than the parental generation, and to comprise more fecund and dispersive females [35]. Additionally, laboratory evidence has shown that the eggs of the oriental fruit moth can tolerate fairly high temperatures, up to 40.5 °C [7]. Since our main objective was to explore the general trend and relationship between the temperature and population of *G. molesta*, the model with temperature predictors reflected the positive effect of temperature during the monitoring period without considering the effect of extreme high temperatures.

We hypothesized that although extreme high temperatures may have affected some individuals in some populations in some sites of the whole population which we analyzed, the general trend cannot be changed at the large scale and in the long term. For example, species with a large geographical range tend to be less affected by temperature [1], while species with a smaller geographical range may be more easily affected by global warming. Insect pests are likely to become more abundant as a result of climate change, whereas biodiversity and conservation are generally being threatened [6], which is possibly due to the small geographical ranges of such pests. An increase in the mean temperature is more detrimental to low-latitude species, and an increase in seasonal fluctuations is more detrimental to high-latitude species [28]. Arthropods adapted for cooler conditions are likely to face the strongest negative effects of global warming during the cooler seasons [36]. Additionally, extreme air temperatures may not represent the climate conditions of the microhabitats encountered by *G. molesta*, because the effects of temperature can be modified by adaptation or by environmental factors [34,37,38]. The thermal safety margin (the difference between the thermal limit and temperature) was greatly overestimated when air temperature or intact leaf temperature was erroneously used [39]. Therefore, the temperature limit in the real world, especially for populations of pests such as *G. molesta*, may be very different from the results determined in the lab. In fact, when a single factor is investigated and interactions with other environmental parameters are not considered, the response of a particular pest species is difficult to determine, because the pest may respond differently during various life stages [6]. Global warming may have a negative impact on insect pests due to the increasing frequency of high temperature extremes; for example, temperate species may be more vulnerable to heat waves than previously thought [39]. However, a meta-analysis found that global warming is expected to be beneficial for major pests; in all but 2 of the 31 globally important phytophagous insect pest species (*G. molesta* not included), the ambient air temperatures moved toward the optimum temperatures for developing life stages [11]. Insect pests may evolve rapidly in response to current rates of global warming [11], owing to the fact that insects are ectothermic creatures with relatively short generation times, and could, therefore, be sensitive to climate change at the population level over short time periods [16]. We hypothesized that major pests may evolve more quickly due to their larger populations, which means more chances of adaptation, and the competition between pests through the filter of human activity may generate superbugs in the agroecosystem.

Although similar works on *G. molesta* on a large scale are currently lacking, some laboratory experiments do partly support our ideas. For example, heat stress negatively affected the fecundity of *G. molesta*, but increased adult heat resistance and adult longevity, both of which have implications for seasonal adaptation as well as changes in dynamics under climate warming [27]. The female *G. molesta* adults could tolerate higher temperatures than

the males through Heat shock protein (Hsp) genes, and the effective induction temperature in females was also lower than that in males [40]. The fitness of some insects may benefit from a higher expression of chaperon genes after mild stress. These benefits would appear in the form of higher fecundity and longer lifespan, as a carry-over effect, and mild thermal stress can also change genetic expression and later boost *G. molesta* adult fitness [41]. Furthermore, *G. molesta* could maximize its fitness by selecting a thermally optimal environment for its offspring, supporting the optimal oviposition theory [37]. The above research may also partly explain why the phenology of *G. molesta* exhibits a strong variation, which could potentially be attributed to regional environmental conditions [42]. Climate change affects the population dynamics of insects in different ways, and thus changes the interspecific interactions [33]. In the future, it will generate competition between herbivores in the agroecosystem. The winners of such competitions may become pests that cause damage and need to be controlled. However, theories or data on how temperature influences intraspecific competition are currently lacking, because when competition is strongest, at temperatures optimal for reproduction, the interaction between temperature and competition leads to more complex dynamics than when competition is independent of temperature [43]. Temperature is an important factor influencing competition within communities of species that utilize the same resources [44]. It seems highly improbable that all members of a community will respond in the same way to climate change [8], and this may lead to competition that favors the major pests. The potential changes in the intrinsic population growth rate will depend on the interaction between mean temperature and thermal variability; the net effect of this interaction could be synergistic or antagonistic [45], and for major pests, it may be more synergistic.

Temperature does not act in isolation to influence pest status, and it is important to consider interactions with other variables; rainfall, for example, is also critical to survival [8]. For example, previous studies have shown that the level of precipitation in spring negatively affects the emergence peak of the overwintering generation, and a low temperature reduces the occurrence level of the first generation [26]. The influence of temperature and humidity on the population dynamics of ectotherms may not necessarily be additive, and more complex interactions could be involved [10]. Thus, we assumed that the influence of precipitation could have a similar effect to that of humidity, since they are generally closely connected. However, when considering temperature and precipitation together, sometimes temperature and precipitation did not appear to have a significant effect on most measures of total herbivore damage, indicating a small insect population [46]. Strong associations between the geographical extent of severe damage and monthly temperature and precipitation are difficult to confirm [47]. The direct impacts of precipitation have been largely neglected in the current research on climate change [1], one reason possibly being that in experiments conducted under laboratory conditions, precipitation is not as easily simulated as temperature. Therefore, the relationship between precipitation and population size has not been studied as often as that between temperature and population size. Comparing the two models, our results show that when adjusting one predictor and the interaction, it can be found that precipitation may have greater effects on *G. molesta* than temperature. A possible reason for this is that we assumed that precipitation might affect the population differently at certain stages, e.g., the overwintering generation, as mentioned above. In general, precipitation may generate higher humidity in orchards; night precipitation, in particular, may provide a suitable environment for the mating of the moths. Additionally, precipitation may also affect the plant production and ecosystem [48], therefore indirectly affecting the population of *G. molesta*.

At the level of the population or higher, the macro-environmental factor may have a greater effect on pests, and the climate parameters may be easy to access, although the patterns of the physiological limits of ectotherms are better explained by environmental gradients than by macroecological processes [39]. This may eventually explain the mechanism underlying it. The challenge remains to identify the causal relationships and to separate them from other factors which may also influence the observed changes in pest

distribution and prevalence in managed ecosystems [6]. Our analyses used mean air temperature, which has several limitations. First, temperature data that have been averaged across months fail to capture extreme weather events. Second, mean air temperatures do not reflect microclimatic variability. Additionally, the number of generations per year is an important characteristic that affects the abundance of multivoltine species [33], and in temperate regions, most insects have their growth period during the warmer part of the year [10]. Trap catches provide quantitative counts of adult males and are highly effective at detecting low-density populations [49]. Pheromone catches of male moths have frequently been interpreted in a chronological sense, with respect to female activity, oviposition, and egg hatching [50]. Other landscape factors, such as peach varieties, may also affects the trap catches. For example, *G. molesta* preferred to oviposit on one peach cultivar when compared with another [51]. Different peach varieties may also affect the *G. molesta* population through mixed odor with certain variability; however, certain varieties may not affect the host location of *G. molesta*, even though they may be preferred by *G. molesta* under laboratory conditions [52]. In addition, the intrinsic rates of the increase in *G. molesta* for fruit variety and fruit species were different at fruit species-level variability, but not at variety-level variability, and the longevity of male moths was also not different at the variety level [53]. All of these temporal and spatial factors may affect the estimates of the models used in this paper, although we attempted to minimize such problems, partially by using random effects.

Climate determines abundance and distribution, but few solid and parsimonious theoretical frameworks are available to examine the effect of climate on population dynamics [10]. Additionally, our results may provide some support for related theories, which may, somehow, become beneficial to the prediction model for the control of *G. molesta* in China in the near future. However, because of the complex interactions of environmental change and the variability in responses (e.g., the responses of their natural enemies), insects are usually subject to large between-generation and between-year fluctuations in abundance. It is, therefore, very difficult to assign trends in any particular species directly to climate change [8]. Within a climatically favorable range, the pest population may be constrained by the availability of host plants and the abundance of natural enemies [54]. Predator–prey and plant–insect interactions are disrupted when interacting species respond differently to global warming [55].

In conclusion, our results showed that both monthly mean temperature and precipitation had a significant positive correlation with the occurrence of *G. molesta*. Therefore, global warming with higher levels of precipitation may favor *G. molesta*, allowing it to outperform other potential pests at the population level in peach orchards, on a large scale.

**Author Contributions:** Conceptualization, H.L. (Hongchen Li), N.H., Q.P., S.W. and H.L. (Hu Li); data analysis, H.L. (Hongchen Li) and N.H.; investigation, N.H., F.Z., X.G., S.W. and Q.J.; writing: H.L. (Hongchen Li), N.H. and S.W.; funding acquisition: Q.J., F.Z., X.G. and S.W. All authors have read and agreed to the published version of the manuscript.

**Funding:** This study has been funded by the Modern Agro-industry Technology Research System (CARS-30-2-01) in China.

**Institutional Review Board Statement:** Not applicable.

**Informed Consent Statement:** Not applicable.

**Data Availability Statement:** The datasets generated during the study are available from the corresponding author upon request.

**Acknowledgments:** The authors are grateful for the support from CARS. The authors also appreciate the three anonymous reviewers for the valuable suggestions that greatly improved this article.

**Conflicts of Interest:** The authors declare no conflict of interest.

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
