# Peer review of "Climate Change Promotes the Large-Scale Population Growth of Grapholita molesta (Busck) (Lepidoptera: Tortricidae) within Peach Orchards in China"

_agronomy, doi:10.3390/agronomy12122954_

Round 1

Reviewer 1 Report

 The manuscript "Climate Change Promotes the Large-Scale Population Growth of Grapholita molesta (Lepidoptera: Tortricidae) within Peach Orchards in China" addresses a topic as interesting as it is current for world agriculture and horticulture.

The work is solid, written coherently and in a logical manner, and has an impressive database behind it. However, I think the potential of the work suffers because of two things:

- the level of detail of the method

- insufficiently explained interpretation.

 In addition, I consider that a linguistic review is necessary as a series of inaccuracies regarding form (redundancy most often) but also grammatical ones (wrong form of some verbs, inverted positioning of noun and verb in some sentences etc) have been identified - here I admit that it would have helped me if the authors have used the template with numbered lines to be able to exemplify. Below, I detail my observations and suggestions:

 The introduction is sufficiently detailed and provides a correct overview of the problem addressed in the article. However, the objectives of the paper are not sufficiently concise. I suggest the authors avoid using the term "our results" when stating the objectives.

Method and materials is the section where several aspects need to be clarified:

- location of the study. The authors will introduce a map with the location of the 17 experimental areas (desirable at the national level). Alternatively, their coordinates (lat, long) can be indicated in tabular form in the Appendix/Supplementary

- I suggest replacing "Data of G. molescta" with "Sampling of G.molesta"

- a possible serious methodological problem could be the omission of some essential information regarding the method of capturing G. molesta. We do not know whether experimental or commercial pheromonal lures were used; if the same pheromonal product and the same type of trap was used in all areas and in all years. Also, due to the specificity of G. molesta in showing preferences for certain peach varieties, a list of all varieties where pheromone traps have been placed will be compiled. Authors will need to provide details of the control of the pest in the experimental plots knowing that these treatments may affect the populations and number of generations of the pest.

 Results:

- probably Chisq could be replaced by χ2, everywhere in the text

- In the phrase "Both values ​​indicate that monthly mean temperature and monthly mean precipitation (night) had a significant positive effect on G. molesta (Figure 2), with the interaction showing a slightly negative but significant effect (−0.012) (Chisq = 6.412 , p < 0.05, df = 1)" the authors will explain in more detail "the significant positive effect of temp and precipitation" but "the interaction slightly negative".

- Also here: "Both values ​​indicate positive effects, although the interaction showed a slightly negative significant effect (−0.014) (Chisq = 8.252, p < 0.01, df = 1)."

 In  the Discussions section, all aspects suggested in Materials and method will be included (bait, traps, peach varieties, etc.). Also the paragraph "Figure 3 also shows that the effect of monthly mean temperature (indicated by the slopes) on G. molesta within the lower temperature range (slope = 0.6352, t = 5.771, p < 0.001, df = 61) was greater than within the higher temperature range (slope = −0.1149, t = −1.112, p = 0.26778, df = 147).” will be moved to the results.

Reviewer 2 Report

This topic is very interesting and is focused on a peach-producing region in China, which makes it relevant. However, its content can be greatly improved. Some points of improvement are:

The introduction and discussion can be enhanced by using literature directly related to the topic of the paper and the species in question. Using the terms "Grapholita molesta" and "climatic change" in the google search engine, 1,300 references were found and in the entire document I do not see references or comparisons with any of these works!

In Material and Methods: For those of us unfamiliar with the regions of China, it would be very helpful to have a map showing the experimental sites. What percentage of peach total production do the study areas represent? Information is required on management programs in the study areas that could affect the results of the study?. In Figure 1 it is not clear what each of the colors used represents.

In Data analysis section: I recommend listing what data they refer were analyzed. The authors writhe “more complicated models with non-normal error distributions failed to better explain the data structure” Please give some examples of these models.

In the results section they indicate “We found that the number of generations of G. molesta in China in-creased from north to south along the latitude” they could be very careful as Latitude by itself is not a determining factor, this determines the temperature and precipitation. Surely, at different latitudes with different altitudes, similar climatic conditions could occur between two sites that determine the same number of generations. I suggest making a clarification of this type in the text.

In Figure 3 It is not clear to me if the graphed data is from all the localities or from one in particular

In Discussion section. The authors do not contrast their results with those of other studies on G. molesta. They make many generalizations, but their results must be focused on the study area. I do not see any paragraph that reinforces or contradicts what there is about the literature of this species

The authors write: ”This result may indicate that G. molesta is not so sensitive to the maximum temperature at the population level at a large scale, or it may mean that G. molesta can endure the extreme temperatures at this stage” This assertion could be valid under the conditions of the study areas, but it should not be generalized. Also, none of the localities in the study area has a maximum temperature over 25 °C which is the optimal temperature for this species (Figure 1), Therefore, that statement is not valid for other areas.

Finally, the authors point out in their introduction that the study was carried out in the main peach-producing area in China, but in the discussion they never gave elements of how this information is useful for this area.

Reviewer 3 Report

All suggestions are in the attached file.
